# Characterization of Copper–Graphite Composites Fabricated via Electrochemical Deposition and Spark Plasma Sintering

**Myunghwan Byun [1],\*, Dongbae Kim [2], Kildong Sung [2], Jaehan Jung [3] , Yo-Seung Song [4], Sangha Park [2],\* and Injoon Son [5],\***

[1]  Department of Advanced Materials Engineering, Keimyung University, Daegu 42601, Korea
[2]  Advanced Materials Research Group, Daegu Mechatronics and Materials Institute (DMI), Daegu 42714, Korea
[3]  Department of Materials Science and Engineering, Hongik University, Sejong 30016, Korea
[4]  Department of Materials Engineering, Korea Aerospace University, Goyang 10540, Korea
[5]  Department of Materials Science and Metallurgical Engineering, Kyungpook National University (KNU), Daegu 41566, Korea
\*  Correspondence: myunghbyun@kmu.ac.kr (M.B.); shpark@dmi.re.kr (S.P.); ijson@knu.ac.kr (I.S.); Tel.: +82-53-580-5228 (M.B.); +82-53-608-2131 (S.P.); +82-53-950-5563 (I.S.); Fax: +82-53-580-5165 (M.B.); +82-53-608-2119 (S.P.); +82-53-950-6659 (I.S.)

**Abstract:** In the present study, we have demonstrated a facile and robust way for the fabrication of Cu-graphite composites (CGCs) with spatially-aligned graphite layers. The graphite layers bonded to the copper matrix and the resulting composite structure were entirely characterized. The preferential orientation and angular displacement of the nano-sized graphite fiber reinforcements in the copper matrix were clarified by polarized Raman scattering. Close investigation on the change of the Raman G-peak frequency with the laser excitation power provided us with a manifestation of the structural and electronic properties of the Cu-graphite composites (CGCs) with spatially-distributed graphite phases. High resolution transmission electron microscopy (TEM) observation and Raman analysis revealed that reduced graphite oxide (rGO) phase existed at the CGC interface. This work is highly expected to provide a fundamental way of understanding how a rGO phase can be formed at the Cu-graphite interface, thus finally envisioning usefulness of the CGCs for thermal management materials in electronic applications.

**Keywords:** copper–graphite composites; anisotropic layered structure; spark plasma sintering; Raman spectroscopy

## 1. Introduction

As the trend in power electronics systems moves toward smart and wearable electronics, effective use of thermal management materials is of critical importance due to the strong demand for the enhancement of power densities and miniaturization and weight reduction [1–6]. Of the various thermal management elements, copper (Cu) is among the most widely used materials for various kinds of passive heat exchangers including sinks, spreads, and pipes because of its excellent thermal conduction (~400 W/m·K) and weldability with solders, thereby making it compatible with electronic applications [7]. However, since Cu has a relatively high density of 8.96 g/cm$^3$ compared to aluminum (2.70 g/cm$^3$), its use can be restricted in applications such as the heat dissipation of electronics, automobile and aerospace electronics, of which space and weight is a strict requirement [3,8,9]. Therefore, fabricating and developing a new heat management materials system

with lower density and good weldability still remains a big challenge in future electronic cooling. Recently, as a promising alternative, carbon allotropes such as diamond [10], natural graphite [11], synthetic graphite [12,13], carbon nanotubes/fiber/flakes [14], and graphene [15–18] have attracted great attention. However, when a polymeric thermal adhesive is applied for passively exchanging the heat at the interfacial region between the heat source and the carbon-based heat dissipater, undesired thermal stress and warpage in electronic components possibly takes place because of discrepancy in thermal expansion coefficients [3,19,20]. Another presumable issue in the solely used carbon-based materials is weak bonding with solders, thus leading to ascending thermal resistance [3,19,20]. To amend the aforementioned issues, effort to combine Cu with carbon-based materials have been made to enhance thermal management capability and concurrently reduce the density of Cu [21,22]. Actually, recent studies have successfully shown a variety of routes to fabrication of Cu-matrix composites by incorporating the carbon allotropes into a Cu matrix such as hot pressing [22], vacuum pressure infiltration [21], electro- [3,23,24] and electroless plating [25], chemical vapor deposition [26], spark plasma sintering [27], etc. Unfortunately, most studies have focused on improvement of thermal conductivity, reduction of Cu density and the distribution of carbon-based reinforcements, rather than characterization of the interfacial region between the Cu matrix and the carbon allotropes as reinforcements.

In the present study, we have demonstrated a facile and robust way for the fabrication of Cu-graphite composites (CGCs) with spatially-positioned graphite phases. The graphite layers bonded to the copper matrix and the resulting composite structure were entirely characterized. The preferential orientation and angular displacement of the nano-sized graphite fiber reinforcements in the copper matrix were clarified by polarized Raman scattering. Close investigation of the change of the Raman *G*-peak frequency with the laser excitation power provided us with a manifestation of the structural and electronic properties of the Cu-graphite composites (CGCs) with spatially-distributed graphite phases. Here, we compared the thermal characteristics of *G*-peak shifts and strengths due to thermal reduction at the copper, graphite, and graphene interfaces in the composites.

## 2. Experimental Procedure

### 2.1. Preparation of Copper–Graphite Composite Materials

Graphite (fiber type, density of ~2.2 g/cm$^3$, Qingdao Krofmuehl Graphite Co., Ltd., Pingdu, China) with a size discrepancy ranging from 100 to 120 μm was selected as the reinforcement material. Prior to electroless plating of Cu on the whole surface of the graphite fiber, the graphite fibers were thermally handled at an elevated temperature of 380 °C in air for 60 min to activate the surface and then the samples were ultra-sonicated in acetic acid ($CH_3CO_2H$). The Cu coating on the graphite fibers was conducted in an electroless plating bath containing an aqueous solution of 70 wt.% cupric sulfate pentahydrate ($CuSO_4 \cdot 5H_2O$) and 10 wt.% formaldehyde (HCHO) at 45 °C with pH values of 8–11 (tuned with a varying content of NaOH). The coating thickness of Cu on graphite fibers was determined with the graphite fraction added to the electroless plating bath. After vigorously rinsing the samples with distilled water, they were dried in a vacuum oven at 60 °C. Finally, the graphite fibers coated with Cu of 2–3 μm were obtained.

### 2.2. Spark Plasma Sintering

Consolidation of copper-coated graphite samples was performed as follows: First, the samples were loaded into a rectangular graphite die (inner diameter of 40 mm × 40 mm) and then thermally treated by using spark plasma sintering (SPS-3.20MK-V, Dr. Sinter, Sumitomo Coal Mining Co., Ltd., Saitama, Japan) under a controlled condition of pressure ~50 MPa and temperature ~920 °C, thereby forming the copper–graphite composites (CGCs).

*2.3. Characterization of Copper–Graphite Composites*

A Raman spectroscopy (InVia Reflex, Renishaw) was applied for characterizing the chemical structure of the carbon in the composites. A characteristic laser wavelength of 514 nm was used to irradiate the surface of the composite material. The interfacial area was carefully observed by using field-emission scanning electron microscopy (FE-SEM; JSM-7900F, JEOL) and transmission electron microscopy (TEM; JEM-2100F, JEOL). The chemical composition of the copper-graphite interface in the copper-graphite composite sintered body was intensively investigated by X-ray photoelectron spectroscopy (XPS; ESCALAB 250).

## 3. Results and Discussion

The graphite fibers were coated with Cu through an electroless plating process, thus forming the CGC powders with a volume ratio of 30 wt.% graphite and 70 wt.% Cu as shown in Figure 1a,b. Thickness of the deposited Cu was observed to be within the range of 2 to 3 μm. Subsequently, these composite powders were consolidated during the spark plasma sintering (SPS) process as displayed in Figure 1c,d. Cross-sectional images of the sintered CGCs uncovered that the graphite powders were layered in the Cu matrix, representing anisotropy in the mixed phase. The graphite layers (black colored region) were spatially distributed in the Cu matrix (white colored region), thus aligning in the plane direction perpendicular to the pressing direction during the sintering process. To clearly identify interfacial reconstruction in between the Cu matrix and the graphite oxide layers and, more importantly, to rationalize the structure formation driven during the SPS process, cross-sectional TEM observations were performed. Such these significant structural changes were made from pristine graphite to graphite oxide (GO), and to the reduced graphite oxide (rGO), which took place during the SPS process. Previous studies of rGO phases formed in the Cu matrix have focused on discussing mechanical properties dominated by the interface strength between Cu and graphene and the graphene dispersion in the composite [28–30]. It is worth noting that the copper–graphite interface with a well-defined thickness of the graphite oxide layers ranging from 8 to 10 nm were formed as obviously shown in Figure 2. Figure 3 shows the XPS results for the chemical composition and ionization etch depth at the copper–graphite composite interface. The Cu and $Cu_2O$ phases, and the composition of carbon, oxygen, and copper compounds was confirmed to be 10 nm from the depth profile of 40 nm on the surface. It could be seen that the composition and quantitative changes in carbon, oxygen, and copper compounds were found to be 20 nm in the depth direction from the surface of the composite material. The amount of carbon and oxygen atoms decreased with distance from the surface, while the pure copper composition increased in weight. This strongly reflected that the rGO and GO interfacial layers formed the anisotropic composite structure [31–33].

Structural changes of the CGCs during the SPS process were further investigated using Raman spectroscopy. The Raman spectrum of the CGC powder showed a *G* band shift at 1581 cm$^{-1}$ corresponding to the primary scattering of graphite, where the intensity was higher than that of the *D* band as displayed in Figure 4. The Raman spectrum of the interfacial graphite and Cu after the SPS process showed a weak intensity peak at 1594 cm$^{-1}$, as well as the *G* band at 1581 cm$^{-1}$. In addition, the intensity of the *D* band at 1363 cm$^{-1}$ increased remarkably. These observations were due to the co-existence of the graphite oxide and graphite phases; hence, we reached the conclusion that the graphite surface was highly expected to be oxidized during the electroless plating and the SPS processes. Furthermore, the increased intensities in *G* and *D* bands (1594 and 1352 cm$^{-1}$, respectively) originated from the formation of GO [34,35]. Since GO and pristine graphite are the different allotropes of carbon, the graphite oxide surface can also be readily bonded to an oxygen-containing group, thereby resulting in ascending distance between the carbon atoms. Therefore, we concluded that the weak peak at 1594 cm$^{-1}$ observed in the Raman spectrum of the CGC material interface turned up after the SPS process [8,36].

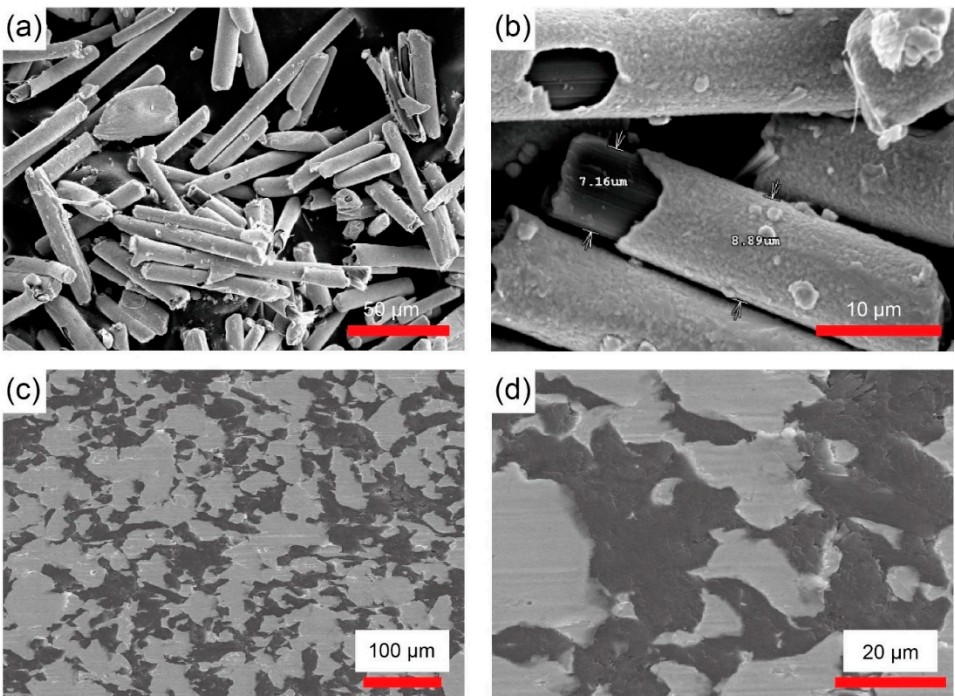

**Figure 1.** Field emission electron microscopy (FE-SEM) images of Cu-coated graphite fibers (**a**,**b**) and Cu-graphite composites after the spark plasma sintering process (**c**,**d**).

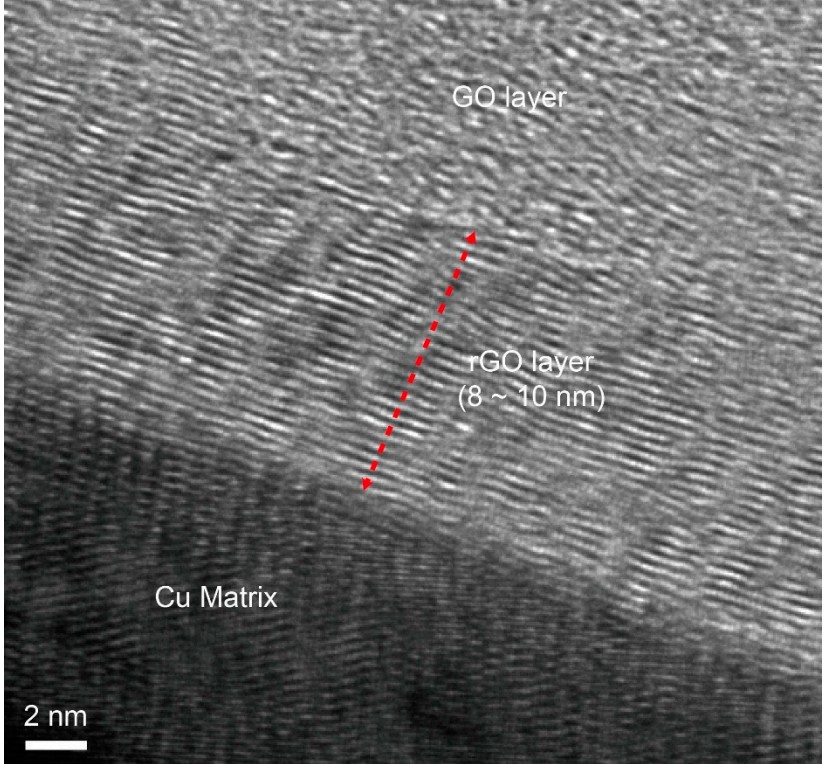

**Figure 2.** High-resolution transmission electron microscopy image of the interface between the copper and graphite composite.

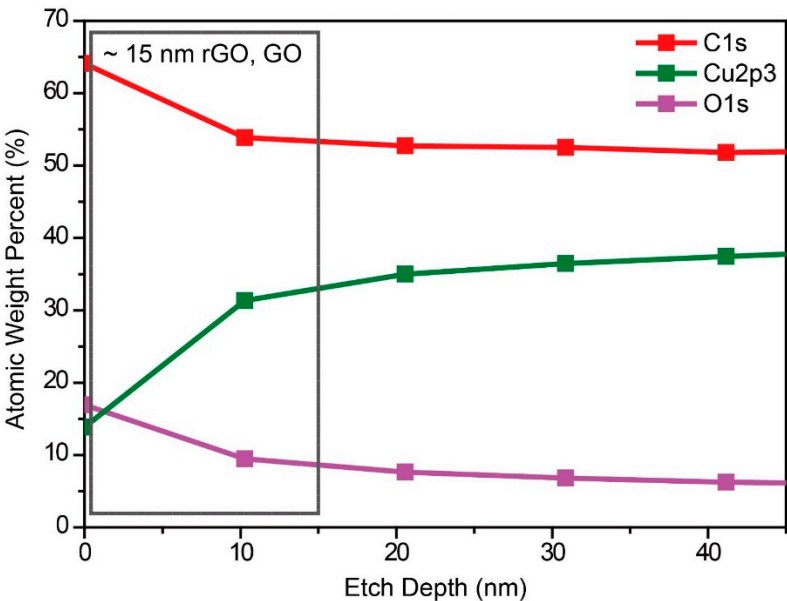

**Figure 3.** X-ray photoelectron spectroscopy of the interface between the copper and graphite composite.

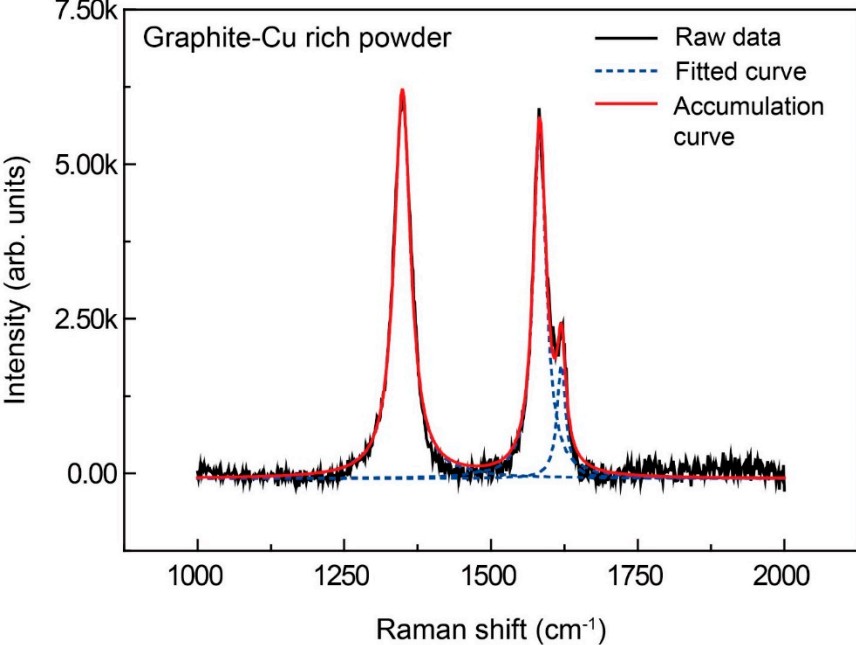

**Figure 4.** Comparison of Raman spectra for the Cu-coated graphite powders, copper–graphite composite interface, and Cu surface.

For further clarification, Raman mapping was performed over the interfacial positions of the copper–graphite interface equivalent to the graphite oxide, reduced graphite powder, and copper–graphite sintered body as clearly marked in Figure 5a. Five representative regions indicate $X_1$ (~1 μm), $X_2$ (~6 μm), $X_3$ (~15 μm), and $X_4$ (~28 μm), which corresponded to the distance from the composite interface, respectively. The *D* band indicates the presence of disordered carbons, whereas the *G* band indicates the graphitized carbon. Therefore, peak changes in *D* and *G* bands over five mapping regions reflect the existence probability of the oxygenic functional groups and disordered carbons on the CGC interface after the SPS process. Particularly, Raman spectra measured at $X_3$ (~15 μm) meant a lower presence probability of the oxygenic functional groups and disordered

carbons compared with other three regions. This assumption could be reasonably rationalized by the fact that GO tends to be readily reduced to graphene-like sheets by getting rid of the oxygenic functional groups with the restoration of a π-conjugated structure.

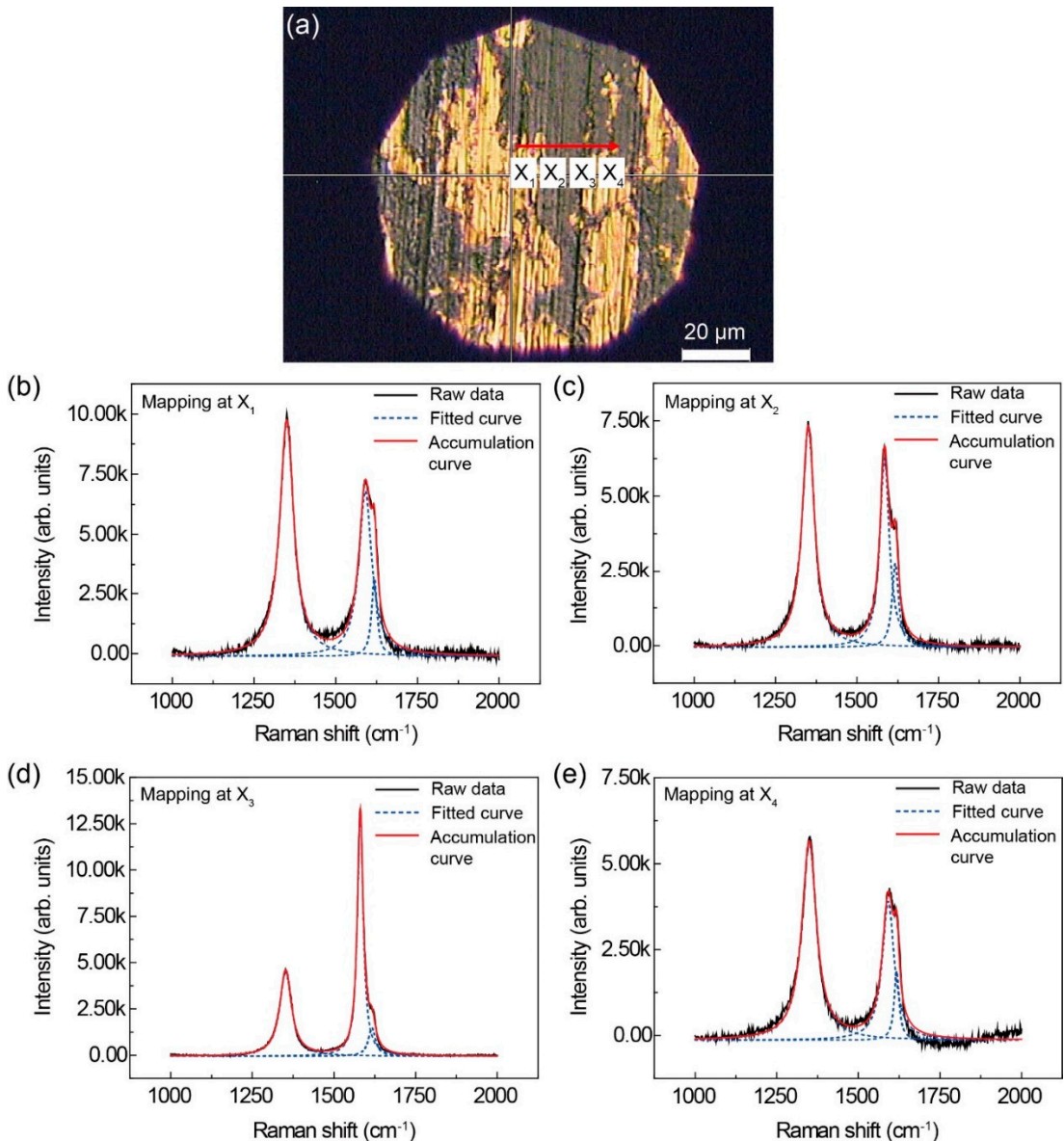

**Figure 5.** Raman mapping was performed over the interfacial positions of the copper–graphite interface equivalent to the graphite oxide, reduced graphite powder, and copper–graphite sintered body. (**a**) A cross-sectional view of the copper-graphite composite visualized by optical microscopy. Raman mapping results for five representative spots of the Cu-graphite composites interface (**b**) 1, (**c**) 6, (**d**) 15, and (**e**) 28 μm from the composite interface.

The Raman shift (cm⁻¹) and intensity ratio ($I_D/I_{G1}$) were introduced to firmly confirm the presence probability of rGO at the composite interface as plotted in Figure 6. As a result, we concluded that rGO was formed at the copper–graphite interface [37,38]. In the case of $I_D/I_{G1}$, analysis of rGO in the copper–graphite interface could not be easily handled because of inconsistence of the values for graphite, GO, and rGO with the mapping results. Characteristic peaks in the Raman spectra varied as the GO underwent chemical reduction. The peak around 1581 cm⁻¹ was attributed to first-order scattering of phonons of *sp*2 carbon atoms, which is generally labeled as the *G* peak; similarly,

a breathing mode of k-point photons of symmetry around 1348 cm$^{-1}$ is referred to as the *D* peak. The $I_D/I_{G1}$ ratio is designated as the degree of disorder, such as that due to defects, ripples, and edges. Figure 6 shows typical analytical data for graphite, GO, and the chemically rGO surface. The $I_D/I_{G1}$ ratios were found to be 0.02, 0.7, and 0.87, respectively. The higher ratio of *D* to *G* bands strongly imply the higher presence probability of oxygenic functional groups and disordered carbons.

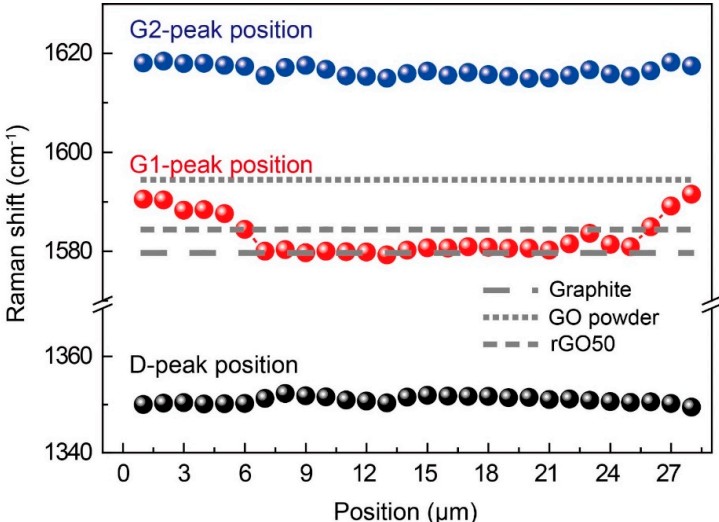

**Figure 6.** Comparison of the *G*2-, *G*1-, and *D*-peak positions and Raman shift (cm$^{-1}$) at the composite interface.

Finally, we investigated the thermal conductivity, conductivity, specific heat of the CGC along the through- and in-plane directions as displayed in Figure 7. All of the thermal properties of the CGC showed an obvious anisotropy, thus leading to 4 to 5 times higher thermal properties in the in-plane direction compared to the through-plane direction. These different tendencies along the through- and in-plane directions were predicted by applying the Hatta Taya method, which mainly assumes that the reinforcements are primarily oriented thin portions homogeneously dispersed in the matrix [39]. In an anisotropic structure, the phonon velocity can be varied along the through- and in-plane directions. However, the interfacial region with micrometer scale roughness could possibly take an effective crystallographic direction for energy propagation across the interface.

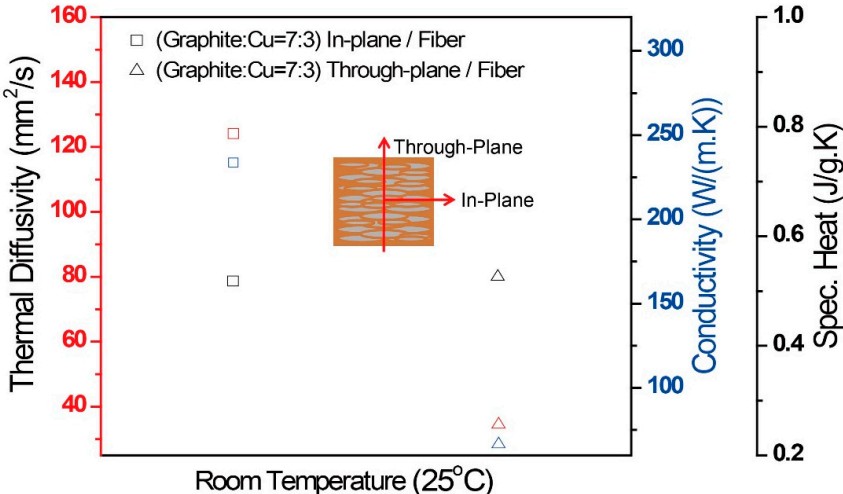

**Figure 7.** Thermal properties of the CGC along the through- and in-plane directions. Inset schematic represents both through- and in-plane directions of the CGC.

## 4. Conclusions

In summary, we demonstrated a facile and robust strategy to fabricate Cu-graphite composites (CGCs) with spatially-aligned, anisotropic layered structures through electroless deposition of graphite reinforcements with Cu and subsequent spark plasma sintering (SPS). On consolidation of the CGCs during the SPS process, rGO was formed at an interfacial region between the Cu matrix and the graphite layers. The formation of unprecedented rGO phases were intensively characterized by FE-SEM, TEM, XPS, and Raman spectroscopy, thus proving the presence probability of rGO phase experimentally. High resolution TEM observation and Raman analysis revealed that rGO phase existed at the CGC interface. This work is highly expected to provide a fundamental way of understanding how rGO phase can be formed at the Cu-graphite interface, thus finally envisioning usefulness of the CGCs for thermal management materials in electronic applications.

**Author Contributions:** M.B., and S.P. designed research; D.K., K.S., J.J., Y.-S.S., and S.P. performed research; M.B., and S.P. analyzed data; and S.P., I.S., and M.B. wrote the paper.

**Funding:** This research was supported by the Keimyung University Research Grant of 2019.

**Conflicts of Interest:** The authors declare no conflicts of interest.

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
