# Peer review of "Characterization of Copper–Graphite Composites Fabricated via Electrochemical Deposition and Spark Plasma Sintering"

_applsci, doi:10.3390/app9142853_

Round 1
Reviewer 1 Report
Thanks for this manuscript.
The authors present a complete study to check the interface between graphite and copper. The results are meaningful and the complete.
Authors studied the formation rGO phases. Has these phases also been identified in other publications? if yes, please compare the findings of other publications.
Author Response
The authors fully appreciate the reviewer's comments.
Please see the attached file including revisions.

Reviewer 2 Report
This study demonstrates a new process for fabricating the Cu-graphite composites and following by the characterization of them. This is an important research field and the paper presents an interesting contribution for better understanding the properties of the Cu-graphite composites. An aspect should be clarified in a revised manuscript before the paper can be considered for publication:
1. The discussions of the thermal behavior or properties should be given as the paper emphasizes on the potential applications to new heat management materials by their investigated composites.
It is felt by this reviewer that by addressing the above issue, the paper would be suitable for publication, therefore accept with major revision is recommended.
Author Response

(The authors gave the same response as above.)

Round 2
Reviewer 2 Report
Accept for publication. Thank you for the reply.